# Plasma Lipidomics Reveals Lipid Signatures of Early Pregnancy in Mares

**DOI:** 10.3390/ijms252011073

**Published:** 2024-10-15

**Authors:** Tharangani R. W. Perera, Elizabeth G. Bromfield, Zamira Gibb, Brett Nixon, Alecia R. Sheridan, Thusitha Rupasinghe, David A. Skerrett-Byrne, Aleona Swegen

**Affiliations:** 1Priority Research Centre for Reproductive Science, School of Environmental and Life Sciences, College of Engineering, Science and Environment, The University of Newcastle, Callaghan 2308, Australiadavid.skerrett-byrne@helmholtz-munich.de (D.A.S.-B.); 2Infertility and Reproduction Research Program, Hunter Medical Research Institute, New Lambton Heights 2305, Australia; 3School of BioSciences, Bio21 Institute, The University of Melbourne, Parkville 3052, Australia; 4SCIEX, Mount Waverley 3149, Australia; 5Institute of Experimental Genetics, Helmholtz Zentrum München, German Research Center for Environmental Health, D-85764 Neuherberg, Germany; 6German Center for Diabetes Research (DZD), D-85764 Neuherberg, Germany

**Keywords:** equine early pregnancy, lipidomics, cholesterol, bile acids, phosphatidylethanolamine, biomarkers

## Abstract

Understanding the systemic biochemistry of early pregnancy in the mare is essential for developing new diagnostics and identifying causes for pregnancy loss. This study aimed to elucidate the dynamic lipidomic changes occurring during the initial stages of equine pregnancy, with a specific focus on days 7 and 14 post-ovulation. By analysing and comparing the plasma lipid profiles of pregnant and non-pregnant mares, the objective of this study was to identify potential biomarkers for pregnancy and gain insights into the biochemical adaptations essential for supporting maternal recognition of pregnancy and early embryonic development. Employing discovery lipidomics, we analysed plasma samples from pregnant and non-pregnant mares on days 7 and 14 post-conception using the SCIEX ZenoTOF 7600 system. This high-resolution mass spectrometry approach enabled us to comprehensively profile and compare the lipidomes across these critical early gestational timepoints. Our analysis revealed significant lipidomic alterations between pregnant and non-pregnant mares and between days 7 and 14 of pregnancy. Key findings include the upregulation of bile acids, sphingomyelins, phosphatidylinositols, and triglycerides in pregnant mares. These changes suggest enhanced lipid synthesis and mobilization, likely associated with the embryo’s nutritional requirements and the establishment of embryo–maternal interactions. There were significant differences in lipid metabolism between pregnant and non-pregnant mares, with a notable increase in the sterol lipid BA 24:1;O5 in pregnant mares as early as day 7 of gestation, suggesting it as a sensitive biomarker for early pregnancy detection. Notably, the transition from day 7 to day 14 in pregnant mares is characterized by a shift towards lipids indicative of membrane biosynthesis, signalling activity, and preparation for implantation. The study demonstrates the profound lipidomic shifts that occur in early equine pregnancy, highlighting the critical role of lipid metabolism in supporting embryonic development. These findings provide valuable insights into the metabolic adaptations during these period and potential biomarkers for early pregnancy detection in mares.

## 1. Introduction

Lipid metabolism in early equine pregnancy presents a fascinating area of study, particularly when considering the unique physiological aspects of mares compared to other mammals [1,2]. How early pregnancy impacts circulating lipid levels and their metabolism is currently unknown. Recent advancements in mass spectrometry and lipidomics have opened new avenues for exploring these complex biochemical pathways, providing deeper insights into pregnancy recognition and immunological responses. 

This study focused on the lipid profile comparison between pregnant and non-pregnant mares, specifically at the critical timepoints of day 7 and 14 post-ovulation, encompassing a stage where equine gestation exhibits distinct characteristics. By day 7 following ovulation, the early embryo descends into the uterus from the oviduct, via a peculiar selective process whereby only embryos, and not unfertilised oocytes, are permitted transit [3]. Once here, the conceptus remains uniquely spherical instead of elongating, encased in an acellular glycoprotein capsule, and maintains a pattern of rapid mobility along both horns of the uterus from day 9 until day 16 [4,5]. From around day 10, the conceptus initiates a critical embryo–maternal interaction known as maternal recognition of pregnancy (MRP), signalling its presence to the maternal system and culminating in the retention of the corpus luteum, which secretes progesterone essential to maintenance of pregnancy [6,7]. The biochemical signal released by the embryo to instigate this cascade of events remains to be identified. Whether these interactions are constrained to the reproductive system or have systemic sequelae, and indeed whether they involve alterations in the lipidome, has not yet been examined.

The importance of lipid profile variations in pregnant mares has been increasingly recognized [1]. Research has documented significant alterations in lipid and lipoprotein profiles during pregnancy in mares. For instance, the effects of different nutritional states on plasma lipid concentrations in pregnant mares and their foetuses have been examined by Stammers, et al. [8], revealing that both short-term fasting and prolonged undernutrition can significantly elevate maternal plasma lipid concentrations, which subsequently influence the lipid profile in the foetal circulation. These observations highlight the sensitivity of maternal and foetal lipid metabolism to changes in nutritional intake during late stages of equine pregnancy. Variations in plasma levels of total cholesterol, HDL-col, total protein, and globulins have also been reported, indicating altered lipid metabolism during pregnancy [9]. Furthermore, research by Shcherbatyy, et al. [10] highlighted increased levels of lipid peroxidation products, such as diene conjugates and malondialdehyde, in pregnant mares, emphasizing that oxidative stress is associated with equine gestation. Arfuso, et al. [11] underscored the dynamic shifts in energy expenditure and lipid profiles during pregnancy, suggesting a complex regulation of lipid metabolism in mares. 

The advanced techniques now available for mass spectrometry in lipidomics offer a powerful tool for dissecting these changes at a molecular level. This technology allows for the precise identification and quantification of a broad spectrum of lipids, providing a comprehensive overview of the lipidome [12]. In this study, we employed high-performance liquid chromatography (HPLC) followed by mass spectrometry (MS) analysis on a SCIEX ZenoTOF 7600 system (SCIEX; Concord, Ontario, Canada) with electrospray ionization and data-dependent acquisition in positive ion mode to measure the lipidomic alterations of plasma samples collected at day 7 and 14 post-ovulation, aiming to deepen the understanding of lipid metabolic changes during early equine pregnancy. The implications of our findings could offer novel avenues for diagnostics and therapeutic strategies aimed at enhancing reproductive efficiency and outcomes in the equine species.

## 2. Results 

### 2.1. Day 7 Characterisation

Untargeted lipidomics analysis of mare plasma returned 2062 identified lipids between the non-pregnant (NP) and pregnant (P) included into 7 main lipid categories, such as Fatty acyls [FA], Glycerolipids [GL], Glycerophospholipids [GP], Prenol Lipids [PR], Sphingolipids [SP], Sterol Lipids [ST], and other lipids and into 43 lipid classes. A full list of lipid IDs is presented in Appendix A. A broad examination of these classes revealed that only the sterols lipid class showed a statistically significant difference between P and NP at day 7 (Figure 1A). The most significant contributor to this difference was BA 24:1;O5. Further interrogation at the lipid species level, identified several species showing differential abundance as depicted in the volcano plot in Figure 1B. Notably, in pregnant mares, lipid ions such as BA 24:1;O5, PI 26:4, PG 38:0_38:5, SHexCer 14:2;O2/24:6, and SM 47:9;O2 (FA 22:6) were significantly upregulated (Fold change ≥ 1.5, *p* ≤ 0.05). Conversely, ions such as PI 27:3 and SM 44:7;O2 (FA 28:0) were found in lower abundance (Fold change ≤ 1.5, *p* ≤ 0.05) as shown in Table 1. These findings point to a complex interplay of lipid molecules during early equine gestation, and this is highlighted by the heatmap (Figure 1C) showcasing the difference in abundances within P and NP samples. 

### 2.2. Day 14 Characterisation

For the lipid profiles detected for day 14 of pregnancy, our analysis revealed that while the overall lipid classes did not show significant differences between pregnant and non-pregnant mares (Figure 2A), there is notable divergence in the abundance of certain lipid species (Figure 2B,C). In pregnant mares, we observed an upregulation in a subset of lipid species which may indicate an active alteration in lipid metabolism. These included an increase in sphingomyelins, phosphatidylinositols, triglycerides, and other lipid species (Figure 2B). Conversely, lipid species, including various ceramides, phosphatidylcholines, and phosphatidylethanolamines, were found to be downregulated in pregnant mares (Figure 2B). This differential abundance of specific lipid species highlights the dynamic nature of lipid metabolism during the mid-stages of early equine pregnancy.

### 2.3. Day 7–14 Characterisation

The transition from day 7 to day 14 in equine pregnancy is marked by significant lipidomic alterations (Figure 3A). By comparing the lipid profiles between day 7 and day 14 of pregnancy, our study observed significant differences in the sterols class, particularly BA 24:1;O5 (Figure 3B). Further, upregulation of certain lipid ions on day 7, including TG O-22:1_15:0_9:0 and TG O-17:1_16:0_18:1 and upregulation of lipids such as Cer 12:0;O3/8:0 (2OH) and NAE 23:1 on day 14 were observed. A standout discovery from our lipidomic analysis is the significant elevation of the sterol lipid BA 24:1;O5 in the plasma of pregnant mares as early as day 7 of gestation (Figure 4A). In Figure 4B, we can see the BA levels changes in day 7 P and NP groups compared to the other sterols in the family and a decrease in BA levels from day 7 to day 14 in Figure 4C. On the other hand, a distinct decrease in phosphatidylethanolamine (PE) was observed in pregnant mare plasma (Figure 5A). Furthermore, Figure 5B shows how the PE levels changed compared with the other Glycerolipids in the family. A decrease in PE was observed from day 7 to 14, as shown in Figure 5C.

In light of the alterations observed in the phosphatidylethanolamine (PE) group, we endeavoured to delve deeper into the saturation data. Intriguingly, our analysis revealed a marked increase in the presence of double bonds in the pregnant (P) group compared to the non-pregnant (NP) group (Figure 6A). A comparative analysis of lipid chain lengths in plasma samples from day 7 and day 14 of pregnancy was made for the PE class. Notably, there is a distinct difference in the distribution of chain lengths between the two timepoints (Figure 7A,B). On day 7 (Figure 7A), there was a wider spread and higher frequency of varying chain lengths, suggesting a significant diversity in lipid species. This is characterized by pronounced peaks at certain chain lengths, indicating a higher abundance of specific lipid molecules with those chain lengths. Conversely, the distribution on day 14 (Figure 7B) appears more uniform with less variation in chain length frequency. 

## 3. Discussion

In this study, we have delved into the plasma lipid dynamics during early equine pregnancy, uncovering significant lipidomic alterations between day 7 and 14 of gestation. Our findings underscore the complex metabolic adaptations necessary for successful embryonic development and implantation, highlighting the crucial role of lipid metabolism in supporting both the embryo and the mare. Integrating our study with existing literature on lipid metabolism across various species provides a richer understanding of these processes and their implications for maternal and foetal health.

By day 7 post-ovulation, glycerophospholipids were predominantly downregulated in pregnant versus non-pregnant mare plasma (Figure 1). Several lipid species were upregulated in pregnant mares; these include glycerolipids, sphingolipids, glycerophospholipids, and a single sterol–bile acid BA 24:1;O5. The latter was also consistently upregulated in day 7 pregnant mares when compared against day 14 pregnant mares (Figure 3 and Figure 4). The upregulation of the bile acid BA 24:1;O5 implies robust cholesterol metabolism, potentially facilitating the provision of cholesterol and its metabolites to the embryo. Bile acids, synthesized from cholesterol in the liver, play a crucial role in cholesterol homeostasis by facilitating the digestion and absorption of dietary fats and cholesterol in the intestine. Therefore, the upregulation of the bile acid BA 24:1;O5 suggests an active cholesterol catabolism, which not only implies robust cholesterol metabolism but also indicates the mobilization of cholesterol and its derivatives. This mobilization is essential for supporting the developmental needs of the embryo, as cholesterol is a vital component for cellular membrane synthesis and is a precursor for steroid hormones and other critical developmental molecules [13,14,15,16,17].

At day 14 post-ovulation, pregnant mare plasma exhibits increased levels of several lipid species such as SM and NaGlycer, while the non-pregnant plasma shows increases in PC and PE. The scatter plot further illustrates these differences, highlighting specific lipid species such as cholesterol, ceramide, and PI that are more prominent in pregnant versus non-pregnant mares (Figure 2).

The elevation in sphingolipids, including the sulfatide SHexCer 14:2;O2/24:6 and sphingomyelin SM 47:9;O2 (FA 22:6), could be indicative of the establishment of embryonic cell membranes and signalling processes that govern early developmental events [18]. Sphingolipids, in particular, play a crucial role in cell membrane signalling, which is vital for early developmental events such as implantation and cell differentiation. The observed decrease in lipids such as PE 18:1/18:2 might suggest a reallocation of phospholipids from the maternal system to the embryo. PE is known to be involved in autophagy and cellular proliferation, which are vital during the rapid cell division occurring at this stage [19]. 

Comparing the plasma lipidomes of same mares at two different stages of pregnancy (Figure 3, Figure 4, Figure 5 and Figure 6) revealed the dynamic nature of lipid metabolism during early gestation and suggests that systemic adaptations in lipid metabolism are vital for successful embryonic development. The upregulation of triglycerides and certain glycerophospholipids on day 7, followed by a shift towards different sphingolipids and phospholipids on day 14, may be reflective of the changing nutritional and structural requirements of the embryo as it prepares for the next phase of development. The observed lipid variations likely support critical processes such as embryonic nutrition, membrane formation, and immune modulation [20]. These changes highlight the finely tuned physiological mechanisms that ensure the viability and proper development of the embryo during the early stages of gestation.

The significant presence of BA 24:1;O5 in early pregnancy aligns with findings in mammalian studies where bile acid production is altered during early gestation. Bile acids are not only a reflection of cholesterol metabolism but also function as signalling molecules that can influence systemic metabolism and energy homeostasis [21,22]. The study by Liu, et al. [23] elucidates the critical role of bile acids in the process of steroidogenesis, highlighting their function beyond their traditional roles in digestion and nutrient absorption. Bile acids, particularly under conditions of cholestasis (a state of elevated bile acid levels due to impaired bile flow), are shown to stimulate steroidogenesis in both mouse models and human adrenocortical H295R cells through a specific signalling pathway involving the sphingosine-1-phosphate receptor 2 (S1PR2), extracellular signal-regulated kinase (ERK), and steroidogenic factor 1 (SF-1). This signalling pathway is independent of the previously known bile acid receptors FXR and TGR5, suggesting a novel mechanism by which bile acids influence adrenal gland function. The findings suggest that supraphysiological levels of bile acids, as seen in cholestasis, can increase steroidogenesis, including cortisol production which has significant implications for adrenal gland physiology. This finding is relevant in the context of pregnancy as explored by Hollinshead, et al. [24], where the increased BA levels in day 7 may be facilitating an initial spike in progesterone production critical for pregnancy initiation before levels adjust for maintenance at day 14. The detection of BA 24:1;O5 may thus reflect the heightened metabolic and biosynthetic activities during early pregnancy, including the synthesis of steroid hormones such as progesterone, which, although not differentially present in the serum, plays a critical role in the maintenance of pregnancy. Furthermore, as serum progesterone levels do not exhibit a marked difference between pregnant and non-pregnant mares specifically on day 7 [25], the detection of BA 24:1;O5 could potentially serve as an alternative, sensitive biomarker for early pregnancy diagnosis.

Regarding the distinct decrease in phosphatidylethanolamine (PE) observed in pregnant mare plasma (Figure 5A), our study aligns with known lipid redistributions during gestation. PE plays a vital role in membrane biogenesis and cell division, essential for supporting rapid cellular proliferation during early gestation. Furthermore, PE is implicated in the initiation of autophagy, a process that could be critical in managing the increased metabolic demands placed on the mother’s body during pregnancy [26]. Additionally, involvement of PE in inflammatory responses could have implications for pregnancy maintenance [27]. Aberrant inflammatory signalling is a known factor in pregnancy complications. Therefore, the decreased levels of PE in pregnant mares may also reflect shifts in the inflammatory milieu, potentially contributing to a favourable environment for pregnancy progression (Figure 5C). Thus, the decreased PE levels in pregnant mare plasma may reflect a combination of modulation of inflammatory responses and, potentially, alterations in cell proliferation and autophagy pathways during pregnancy. Figure 5B shows how the PE levels change in comparison with the other Glycerolipids in the family. The integration of our lipidomic data with these findings provides a richer understanding of the complex metabolic adaptations occurring during early equine gestation.

As pregnancy progresses, initial differences in lipid saturation between pregnant (P) and non-pregnant (NP) mare plasma appear to attenuate by day 14. This observation suggests that the early pregnancy phase is characterized by a distinct lipidomic profile, which undergoes a notable transition toward a more stabilized state of lipid metabolism. Specifically, the pronounced differences in lipid saturation and double bond presence observed in the P group relative to the NP control group on day 7 are not as apparent by day 14 (Figure 6A). This may reflect physiological adaptations where the heightened metabolic demands of early pregnancy are met through increased lipid saturation [28], yet by day 14, the demand appears to normalize, potentially indicating an adaptation phase is completed. The importance of lipid fluidity in maintaining membrane functionality for nutrient transport and signalling remains vital throughout pregnancy, but the initial surge in lipid saturation and unsaturation seen in early gestation seems to balance out, suggesting a possible consolidation in lipidomic activity. This trend underscores the dynamic nature of lipid metabolism during the early stages of pregnancy and its possible stabilization as gestation proceeds, which could have implications for embryo–maternal interactions and the overall regulation of inflammatory responses [29] during this critical period of development. The influence of the oestrous cycle on lipid metabolism may further contribute to these differences, as hormonal shifts can significantly impact lipid distribution and function. Although the specific lipid ions identified in our study do not directly match those found by Hughes, et al. [30], the general trends observed, such as changes in lipid profiles during the oestrous cycle and early gestation, are consistent. These patterns underscore the importance of lipid metabolism in early equine pregnancy, particularly in the context of progesterone synthesis, immune modulation, and embryonic development.

## 4. Materials and Methods 

### 4.1. Mare Plasma Sample Collection

Thoroughbred mares aged 3–16 years (*n* = 28) enrolled in a commercial breeding operation were randomly selected from two breeding farms in the Hunter Valley region of New South Wales, Australia. Procedures were approved by the University of Newcastle Animal Care and Ethics Committee (approval number A-2018-804). The animals were considered healthy based on veterinary records, physical examinations, and reproductive tract examinations performed by rectal palpation and ultrasonography. An expert veterinary surgeon used ultrasound scanning to monitor ovarian and uterine activity during estrus. The mares were then bred by live cover one day before, or on the day of, estimated ovulation, and ovulation was confirmed the following day. Mares were checked for pregnancy 12–14 days after ovulation to confirm the pregnancy status by ultrasonographic examination. All mares were maintained in an outdoor paddock with the same diet (predominantly pasture with supplementary feeding) and ad libitum water. Blood samples were collected from the jugular vein of the mares and dispensed into 10 mL EDTA anticoagulant tubes. To maintain the stability of lipids, blood samples were centrifuged (3000× *g*, 10 min, RT) within 1 h of collection (samples were kept on ice). The resultant plasma was stored at −80 °C until further processing. Previous studies have indicated that lipids, particularly lipoproteins, are minimally affected by a delay in centrifugation up to 8 h, demonstrating high resilience to preanalytical handling variations [31]. While metabolites can show significant changes with delayed processing, lipoproteins remain relatively stable, making them robust biomarkers even when immediate processing is not feasible [32]. Our approach aligns with these findings, supporting the reliability of our lipidomic data. Pregnancy maintenance details and foaling dates were also recorded, allowing samples to be categorized as pregnant at day 7 (7P), *n* = 11 or non-pregnant (7NP) *n* = 14 and at day14 pregnant (14P) *n* = 12 or non-pregnant (14NP) *n* = 13. 

### 4.2. Lipid Extraction

Samples were prepared for mass spectrometry runs in batch mode (12 samples at once, 3 samples per each group). Briefly, the extraction solvent was prepared by mixing 15 mL of butanol (BuOH), 15 mL of methanol (MeOH) with 100 µL of Internal Standard (SPLASH® Lipidomix® Mass Spec Standard (Avanti Polar Lipids, 330707; Sigma-Aldrich, Merck Darmstadt, Germany) [33], and 10 mM ammonium acetate (NH4OAc) together, creating a 10 mM extraction solvent. The solution was well vortexed at room temperature. Twenty microliters of plasma were aliquoted into LoBind Eppendorf tubes, and 180 µL of the prepared extraction solvent was added to each plasma sample, the total to be 200 µL. The samples were vortexed for 30 s to ensure thorough mixing and then incubated on a shaker at 2000 rpm on 22 °C for 60 min to facilitate the extraction process. Following the incubation, the samples were centrifuged at 13,000 rpm for 15 min at room temperature to precipitate proteins, DNA, and cellular debris. The supernatant, 150 µL from each sample, was carefully transferred into glass HPLC vials equipped with inserts and Teflon insert caps for subsequent HPLC analysis. To prepare the Pool Biological Quality Control (PBQC), 20 µL from each extraction was mixed to create a pooled sample. The PBQC mix was vortexed thoroughly and aliquoted into separate vials. A PBQC sample was placed to run at regular intervals throughout the analytical run, ideally after every 10 samples, to ensure quality control and consistency in the analysis.

### 4.3. High-Performance Liquid Chromatography (HPLC)—Mass Spectrometry (MS) 

Lipid separation was achieved on a Phenomenex Kinetex C18 column (2.6 µm, 100 Å, 100 × 2.1 mm). A 5 µL sample injection volume was utilized. Mobile phase A consisted of 1 mM sodium acetate in a mixed solvent of 50% water, 30% acetonitrile, and 20% isopropanol. Mobile phase B comprised 10 mM ammonium acetate in 90% isopropanol, 9% acetonitrile, and 1% water. The separation employed a 15 min gradient protocol, transitioning from 10% to 100% B, facilitating the comprehensive resolution of lipid species. Lipid analysis was conducted using a SCIEX ZenoTOF 7600 system (SCIEX; Concord, Ontario, Canada) accessed at the Sydney Mass Spectrometry, a core research facility at the University of Sydney, equipped with a DuoSpray Turbo V (SCIEX; Concord, Ontario, Canada) ion source and an electrospray ionization (ESI) probe. Instrument calibration was maintained using the automated calibrant delivery system (CDS), which calibrated every five samples with an ESI calibration solution. Data-dependent acquisition (DDA) scans were performed in the positive ion mode. This mode was chosen for its efficacy in ionizing lipid molecules, generating positively charged ions for analysis. The system was configured to select the top 10 most abundant ions for fragmentation, enhancing the detection of significant lipid species. Dynamic background subtraction (DBS) with a mass tolerance of 50 mDa was applied to minimize noise and enhance signal clarity. The time-of-flight (TOF) mass spectrometer accumulation time was set at 250 ms, and a collision energy (CE) of 10 V, with a 50 ms accumulation time, was employed for TOF MS/MS acquisitions, facilitating detailed structural elucidation of lipid molecules. 

### 4.4. Data Processing and Analysis

Data processing and analysis were conducted using MS-DIAL software (version 5.1). The software utilized a comprehensive lipid library, encompassing MS/MS fragments generated by collision-induced dissociation (CID) and electron-activated dissociation (EAD). This robust database allowed for precise lipid molecular species identification with 99% confidence using MS/MS fragmentation data, ensuring accurate and reliable lipidomic profiling. Identification confidence levels were assigned as Level 1 [34], based on two different fragmentation techniques, collision-induced dissociation (CID) and electron-activated dissociation (EAD), which provides an additional information of the regio-isomer and double bond positions of the lipid structure. We used the MS-DIAL silico predicted lipid library based on both CID and EAD fragmentations for identification of the lipid species. While this approach provides reliable lipid profiling, exact bond positions remain challenging without additional validation techniques such targeted lipidomics [35]. Raw data were prepared using MS dial software version 4.9.2, and further statistical analysis were carried out using software 5.0 [36]. Basic data handling, if not otherwise stated, was conducted using Microsoft Excel 365 (Version 16.0.4966.1000, Microsoft Corporation, Redmond, WA, USA) and in GraphPad Prism version 10.1 (GraphPad Software; San Diego, CA, USA).

## 5. Conclusions

In summary, our investigation into the lipidomic signatures of early equine pregnancy has unveiled critical distinctions in lipid metabolism between pregnant and non-pregnant mares. Central to our findings is the significant elevation of the sterol lipid BA 24:1;O5 in the plasma of pregnant mares as early as day 7 of gestation, marking it as a potential sensitive biomarker for early pregnancy detection. This is especially notable considering the lack of discernible difference in serum progesterone levels at this stage. Our study highlights the crucial role of sterols, derived from cholesterol metabolism, in supporting pregnancy. These sterols are imperative for a range of biological processes, including energy provision, cellular membrane synthesis, and the production of steroid hormones critical for gestation maintenance. The heightened presence of BA 24:1;O5 correlates with the essential role of cholesterol in the metabolic activities unique to early pregnancy, particularly in the synthesis of progesterone and other steroids. Moreover, our research indicates dynamic shifts in cholesterol synthesis and utilization during early gestation. The observed decrease in BA levels from day 7 to day 14 of gestation suggests a metabolic adaptation, possibly due to the increased demand for cholesterol in steroid hormone biosynthesis, essential for sustaining pregnancy. These findings are consistent with other mammalian studies showing altered bile acid production in early pregnancy, where bile acids serve not just as metabolic byproducts but also as signalling molecules influencing overall metabolism and energy balance. 

In conclusion, our findings indicate that systemic lipidomic alterations can occur as early as day 7 post-ovulation, a significant departure from previous assumptions that such changes manifest later in gestation. This early onset of systemic changes suggests potential implications for developing early pregnancy diagnostics and exploring novel pregnancy-supportive interventions beyond progestins. However, the sample size of our study (*n* = 28) presents a limitation that may affect the broader applicability of the findings. Thus, future research with larger cohorts and targeted lipidomic analyses will be necessary to validate these observations and translate them into actionable clinical or veterinary practices.

## Figures and Tables

**Figure 1 ijms-25-11073-f001:**
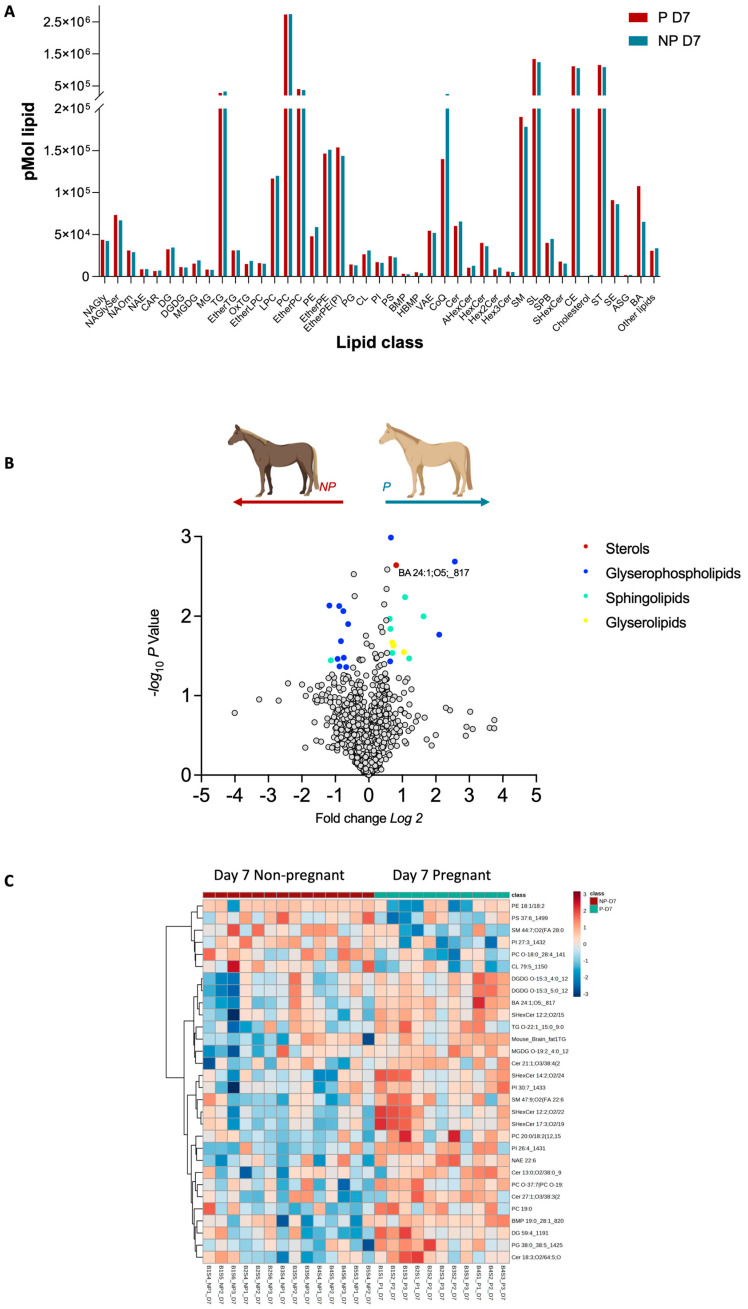
Pregnancy-status-dependent lipidomics characteristics at day 7 post ovulation. (**A**). Distribution of whole lipidome according to their classes. (**B**) Volcano plot depicts the quantitative comparison between the P and NP groups. Coloured dots represent upregulated and downregulated lipids, respectively (fold-change ≥ 1.5 and *p* ≤ 0.05). Gray dots represent lipids that are not differentially expressed between the groups. (**C**) Heat map showing the lipids abundance patterns in pregnant and non-pregnant plasma. Ratio is mapped from red (increase) to blue (decrease) or white (no change); see colour key inset. Each row represents a lipid species, and each column represents a sample.

**Figure 2 ijms-25-11073-f002:**
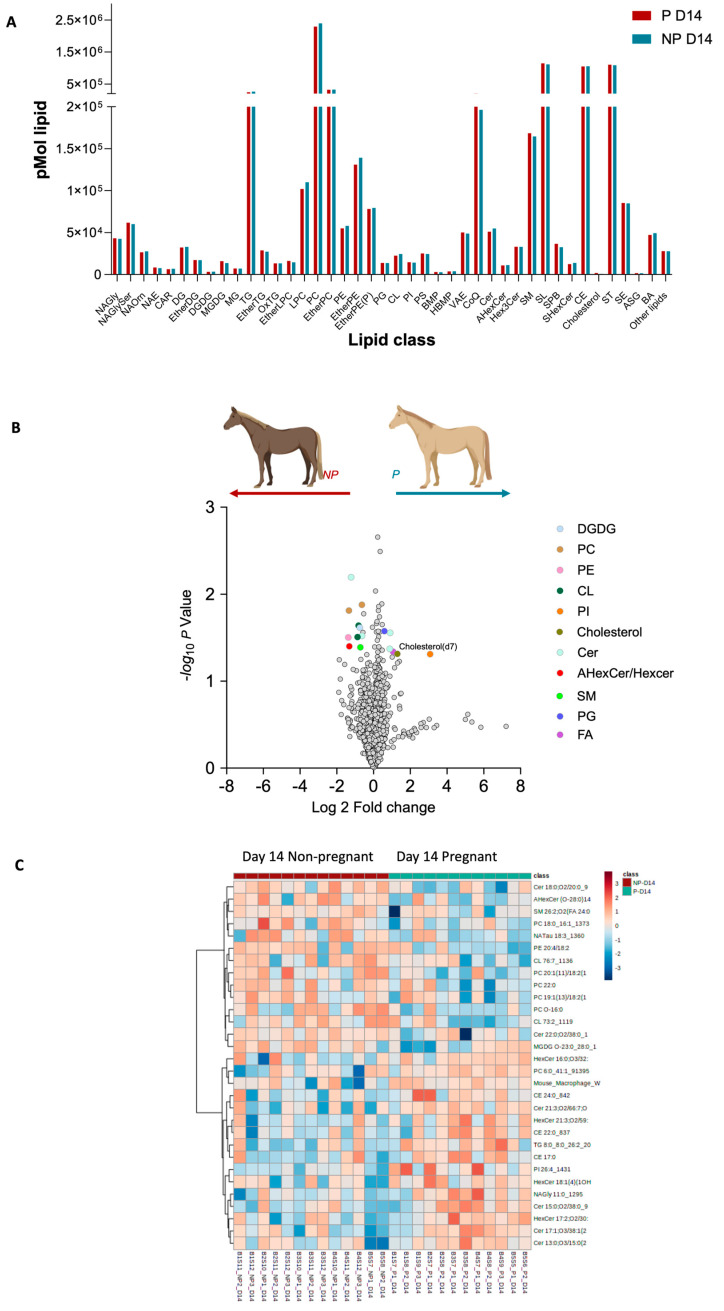
Pregnancy-status-dependent lipidomics characteristics at day 14 post ovulation. (**A**). Distribution of whole lipidome according to their classes. (**B**) Volcano plot depicts the quantitative comparison between the P and NP groups. Coloured dots represent upregulated and downregulated lipids, respectively (fold-change ≥ 1.5 and *p* ≤ 0.05). Gray dots represent lipids that are not differentially expressed between the groups. (**C**) Heat map showing the lipids abundance patterns in pregnant and non-pregnant plasma. Ratio is mapped from red (increase) to blue (decrease) or white (no change); see colour key inset. Each row represents a lipid species, and each column represents a sample.

**Figure 3 ijms-25-11073-f003:**
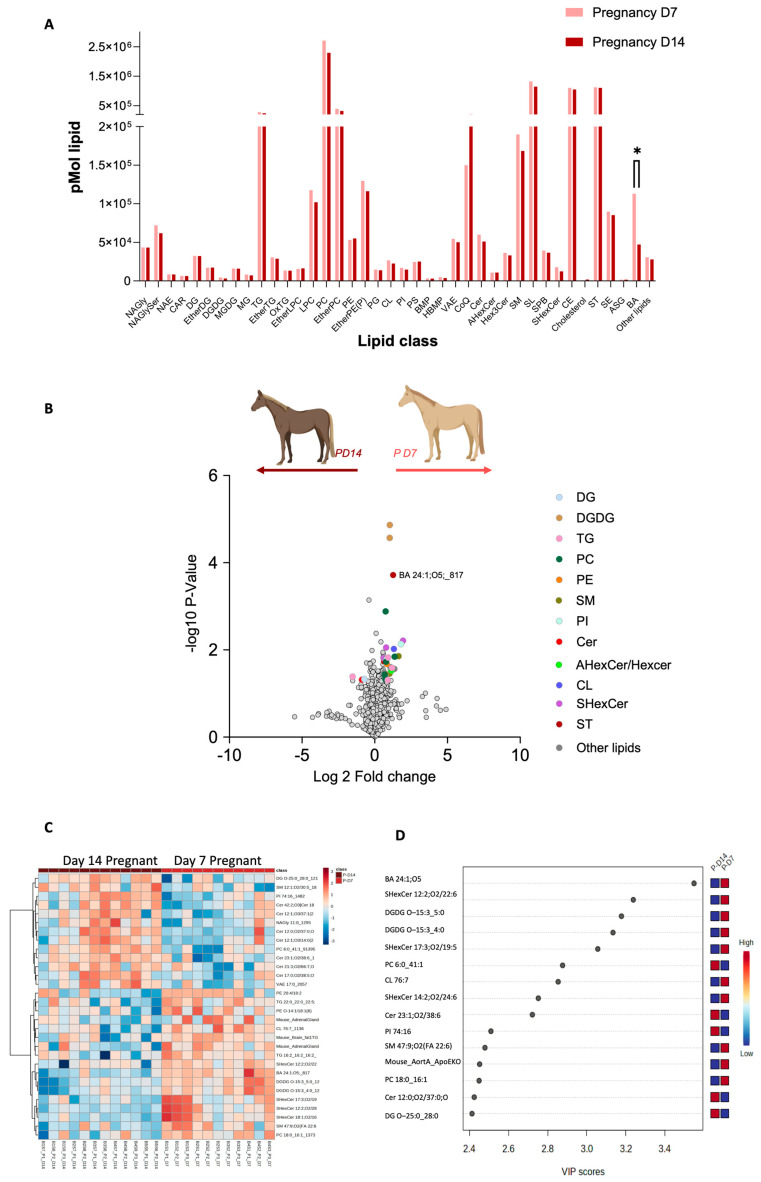
Pregnancy-status-dependent lipidomics characteristics when pregnancy is progressing from day 7 to day 14. (**A**). Distribution of whole lipidome according to lipid classes (* indicates a statistically significant difference (*p ≤* 0.05) between the groups). (**B**) Volcano plot depicts the quantitative comparison between the P and NP groups. Coloured dots represent upregulated and downregulated lipids, respectively (fold-change ≥ 1.5 and *p* ≤ 0.05). Gray dots represent lipids that are not differentially expressed between the groups. (**C**) Heat map showing the lipids abundance patterns in pregnant and non-pregnant plasma. Ratio is mapped from red (increase) to blue (decrease) or white (no change); see colour key inset. Each row represents a lipid species, and each column represents a sample. (**D**) Graph illustrating fifteen lipid species, i.e., chemical shifts, corresponding to assigned lipids with VIP values greater than 1.95 according to PLS-DA results. The coloured scale on the right represents the variation in concentrations of lipids in the P and NP plasma lipids.

**Figure 4 ijms-25-11073-f004:**
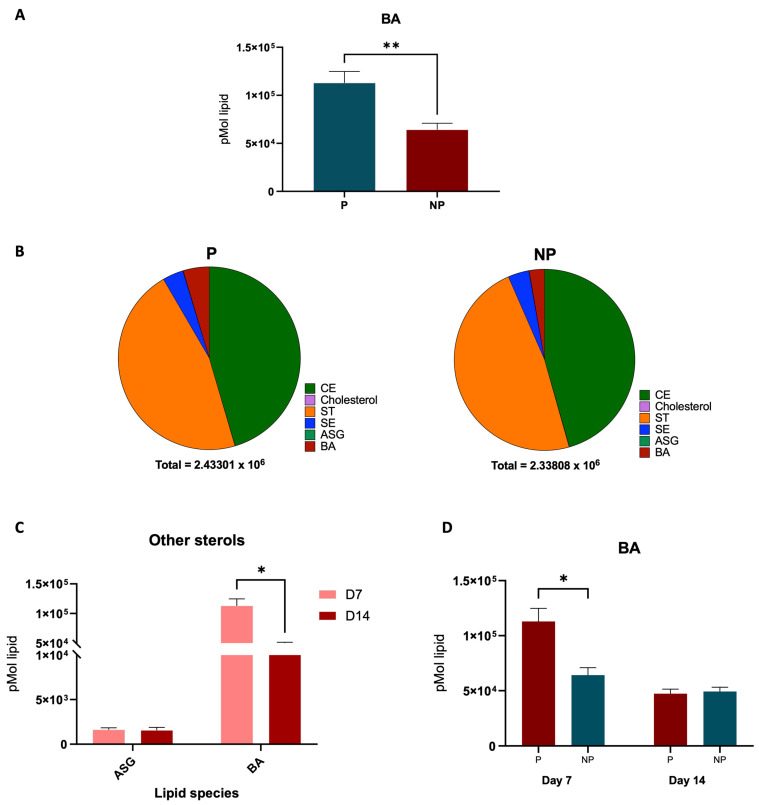
Changes in BA within groups. (**A**) BA difference in P and NP plasma (** indicates a statistically significant difference (*p ≤* 0.01) between the two groups). (**B**) Comparison of BA abundance to the other lipids in the sterol lipid class. (**C**) Occurrence of BA in comparison to the other sterols. (**D**) Graph shows the significant difference in BA when pregnancy progression from day 7 to day 14 (* indicates a statistically significant difference (*p ≤* 0.05) between groups).

**Figure 5 ijms-25-11073-f005:**
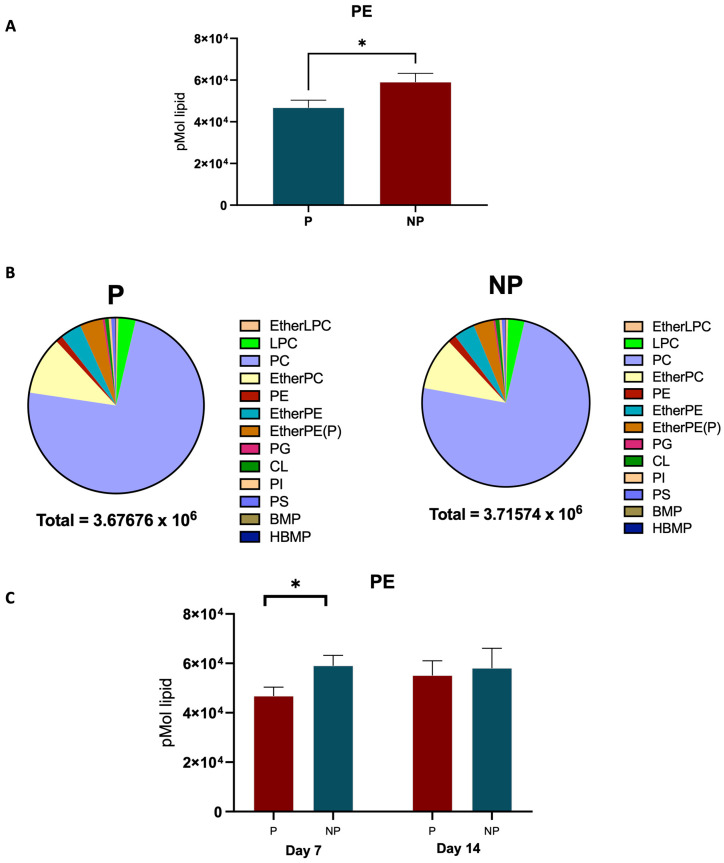
Changes in PE within groups. (**A**) PE difference in P and NP plasma. (**B**) Comparison of PE abundance to the other lipids in the Glycerophospholipid class. (**C**) Graph shows the significant difference in PE when pregnancy progression from day 7 to day 14 (* indicates a statistically significant difference (*p ≤* 0.05) between groups).

**Figure 6 ijms-25-11073-f006:**
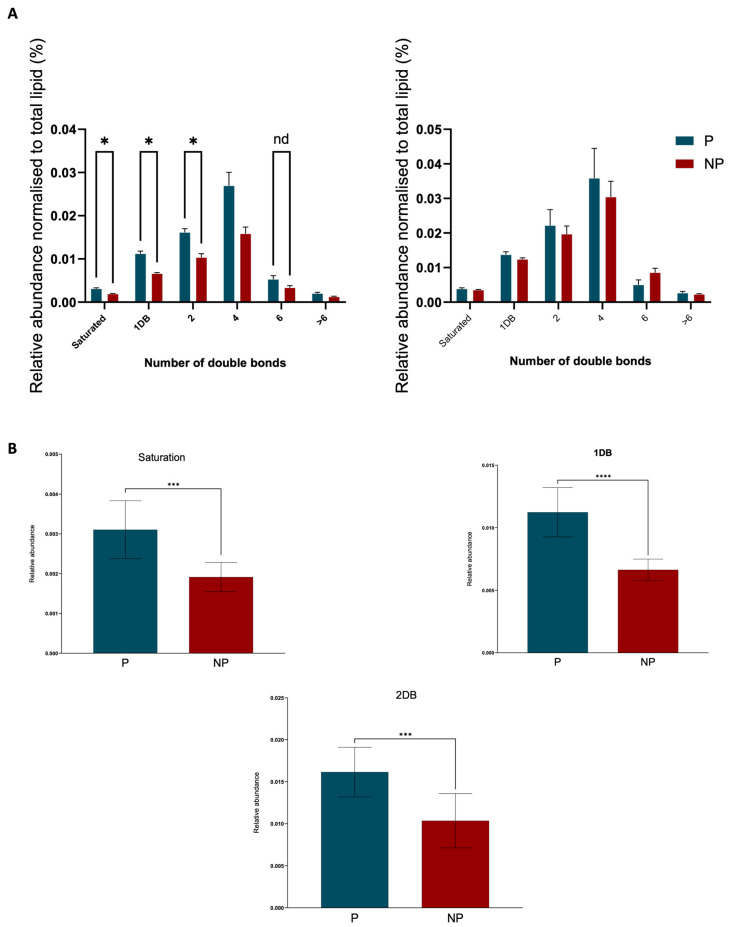
Saturation data at day 7 for PE group. (**A**) Graphs shows the significant difference saturation data when pregnancy progression from day 7 to day 14 (* indicates a statistically significant difference (*p ≤* 0.05) between the groups). (**B**) The significantly different saturation data for 0, 1, and 2 double bonds (*** and **** indicate statistically significant differences between the groups; *p ≤* 0.0002 and *p ≤* 0.0001 respectively).

**Figure 7 ijms-25-11073-f007:**
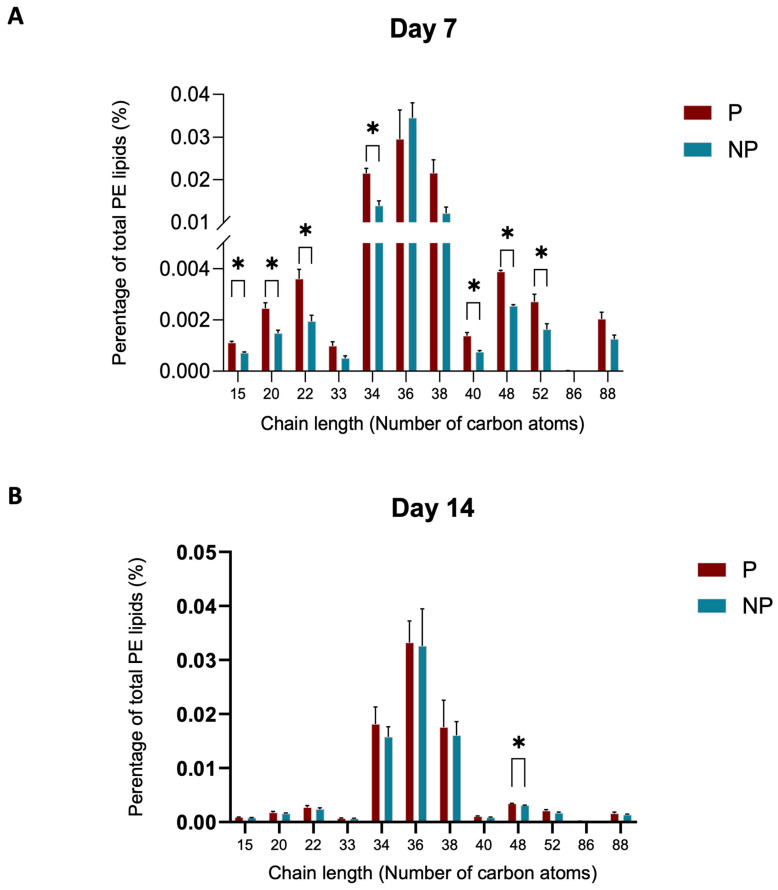
Chain length differences in PE group when pregnancy progresses from day 7–day 14. (**A**) The chain length of the lipid species in PE group at day 7. (**B**) The chain lengths of lipid species in day 14 PE lipids (* indicates a statistically significant difference (*p ≤* 0.05) between the groups).

**Table 1 ijms-25-11073-t001:** Upregulated and downregulated lipids on day 7: pregnant and non-pregnant comparison.

Description	Log_2_Fold Change (P/NP)	*p*-Value
PI 26:4	2.57	0.0023
PG 38:0_38:5	2.10	0.0189
SHexCer 12:2;O2/15:4	1.64	0.0112
SM 47:9;O2 (FA 22:6)	1.21	0.0376
Cer 12:0;O3/36:0 (2OH)	1.09	0.0064
TG O-22:1_15:0_9:0	1.05	0.0315
BA 24:1;O5;	0.82	0.0025
MGDG O-19:2_4:0	0.74	0.0261
TG O-17:1_16:0_18:1	0.74	0.0238
SM 40:6;O2	0.70	0.0321
BMP 19:0_28:1	0.66	0.0011
Cer 15:0;O2/38:0;O	0.65	0.0160
CL 76:0	0.64	0.0411
Cer 21:1;O3/38:4 (2OH)	0.63	0.0119
PC 16:1/18:3	−0.62	0.0140
PC 17:2/18:2	−0.68	0.0486
CL 70:2	−0.74	0.0368
PC O-18:0_28:4	−0.76	0.0096
PC 18:3/18:2	−0.83	0.0229
LPC 18:3/0:0	−0.87	0.0473
PI 27:3_1432	−0.88	0.0083
PC 18:3 (9,12,15)/18:2 (12,15)	−0.93	0.0381
TG 51:0	−1.13	0.0400
PE 18:1/18:2	−1.18	0.0082

## Data Availability

The original contributions presented in the study are included in the article/Appendix A, further inquiries can be directed to the corresponding author.

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
