# Peer review of "Plasma Lipidomics Reveals Lipid Signatures of Early Pregnancy in Mares"

_ijms, 2024, doi:10.3390/ijms252011073_

Round 1
Reviewer 1 Report
Comments and Suggestions for Authors
Review of the Paper: Plasma Lipidomics Reveals Lipid Signatures of Early Pregnancy in Mares
Strengths:
This study offers valuable understanding of the lipidomic alterations that take place in the early stages of equine pregnancy, specifically in relation to pregnant and non-pregnant mares. Robust data on lipid profiles are provided by high-resolution mass spectrometry, an area that has not been extensively studied in the context of early pregnancy.
The study presents a dynamic picture of lipid metabolism during the early stages of gestation by examining lipid alterations at two different time points (days 7 and 14 post-ovulation). The ability to comprehend temporal variations in pregnancy-related lipid profiles is greatly enhanced by this time-resolved method.
The utilization of state-of-the-art lipidomic methods, like data-dependent acquisition HPLC-MS, enhances the accuracy and dependability of the results.
Recommendations:
A sample size of 28 is small to draw strong conclusions, particularly given the known variability in lipid metabolism.
The authors must explicitly recognize this limitation and avoid overstating the significance of their results.
The lack of a more substantial cohort undermines the strength of the conclusions drawn.
While the paper identifies various lipid changes, the authors fail to provide any meaningful biological interpretation of these shifts.
Reporting elevated bile acid BA 24:1;O5 without discussing its specific role in pregnancy and embryonic development leaves the reader with unanswered questions.
The paper severely lacks in translating its findings into actionable clinical or veterinary applications.
The authors need to go beyond merely identifying lipid changes and clearly articulate how these findings can be applied, especially as potential biomarkers for pregnancy diagnostics or reproductive management.
The discussion is notably weak in integrating these findings with the current body of literature on lipid metabolism in pregnancy.
The authors need to significantly expand this section to highlight how their work fits into the broader scientific context.
Overall:
Although the study clarifies the lipidomic alterations that occur during the early stages of equine pregnancy, it falls short in terms of offering a significant biological interpretation and a useful application of its findings. To increase the paper's significance, the authors must address these important problems. The study's value is still restricted in the absence of more thorough analysis, a larger cohort, and a more distinct link to therapeutic relevance.
Author Response
IJMS-3190061
RESPONSE TO REVIEWER COMMENTS
RESPONSE TO REVIEWER #1
Strengths:
This study offers valuable understanding of the lipidomic alterations that take place in the early stages of equine pregnancy, specifically in relation to pregnant and non-pregnant mares. Robust data on lipid profiles are provided by high-resolution mass spectrometry, an area that has not been extensively studied in the context of early pregnancy. The study presents a dynamic picture of lipid metabolism during the early stages of gestation by examining lipid alterations at two different time points (days 7 and 14 post-ovulation). The ability to comprehend temporal variations in pregnancy-related lipid profiles is greatly enhanced by this time-resolved method. The utilization of state-of-the-art lipidomic methods, like data-dependent acquisition HPLC-MS, enhances the accuracy and dependability of the results.
Answer: Thank you for your positive evaluation of our study. We are pleased to hear that the comprehensive analysis of lipidomic alterations during early equine pregnancy was well-received. The use of high-resolution mass spectrometry, including data-dependent acquisition HPLC-MS, was indeed crucial in capturing the dynamic changes in lipid profiles between pregnant and non-pregnant mares. By examining these alterations at two distinct time points, days 7 and 14 post-ovulation, we aimed to provide a clearer understanding of the temporal shifts in lipid metabolism that are essential for early gestation.
One of the major clinical implications of our findings is the identification of systemic changes occurring as early as day 7 post-ovulation. This represents a radical departure from previous assumptions, which typically suggested that such changes occur later in gestation. Detecting these early systemic alterations enhances our understanding of the physiological adaptations during early pregnancy and could potentially lead to improved strategies for early pregnancy diagnosis and management in equine practice.
We appreciate your recognition of the novelty and significance of this approach, as lipidomics in the context of early pregnancy has been underexplored. Our goal was to offer a detailed, time-resolved view of pregnancy-related lipid variations, and we believe that our findings contribute valuable insights into the metabolic adaptations occurring during this critical period. We hope that this study will serve as a foundation for further research in this area and help advance reproductive management strategies in equine practice.
Thank you once again for your thoughtful and encouraging feedback.
- 1. A sample size of 28 is small to draw strong conclusions, particularly given the known variability in lipid metabolism.
Answer: We acknowledge the limitation of the sample size and have revised the manuscript to clearly state this in the limitations section (Conclusion, please see lines 430 – 431). We agree that the variability in lipid metabolism can impact the results, and we have ensured that our conclusions are appropriately cautious, emphasizing the exploratory nature of the study and suggesting that larger cohort studies are necessary for further validation.
- 2. The authors must explicitly recognize this limitation and avoid overstating the significance of their results.
Answer: We appreciate this feedback and have modified the Results and Discussion sections to ensure that the significance of our findings is not overstated (lines 310-315, 365-366, 426-433, 410-411). We have used language that reflects the discovery nature of the findings and suggests potential rather than definitive conclusions.
- The lack of a more substantial cohort undermines the strength of the conclusions drawn.
Answer: Thank you for your comment regarding the cohort size and its impact on the strength of the conclusions drawn. We would like to emphasize that this study was conducted as an exploratory pilot investigation, aimed at assessing whether systemic lipidomic changes could be observed in the early stages of equine pregnancy. Given the small sample size, we recognize that the findings should be interpreted with caution, and the potential for false-positive results cannot be entirely ruled out (lines 432-434).
The primary objective was not to draw definitive conclusions but to generate preliminary data that could guide the design of a more comprehensive study. Encouraged by the initial results, we plan to expand this research with a larger cohort and employ targeted lipidomics to validate the specific lipid alterations identified in this study. This future research will be crucial for confirming the observed trends and ensuring that the findings are robust and generalizable.
We appreciate your feedback and acknowledge the importance of this study as a foundational step towards understanding the systemic lipidomic changes associated with early pregnancy, while being mindful of its limitations.
- 4. While the paper identifies various lipid changes, the authors fail to provide any meaningful biological interpretation of these shifts.
Answer: Thank you for your comment regarding the biological interpretation of the lipid changes identified in our study. We acknowledge that a comprehensive interpretation of all significantly altered lipids was not feasible within the scope of this manuscript. Instead, we concentrated on a select number of interesting lipids that showed the most promise for further investigation, and we have provided detailed explanations for their potential roles in biological pathways.
For example, we have discussed the upregulation of specific bile acids and their involvement in cholesterol metabolism and steroidogenesis, which are crucial during early pregnancy (lines 303-315). Similarly, we explored the significance of elevated triglycerides and phospholipids in cell signalling processes associated with embryo-maternal communication (lines 334-341). These interpretations are included in the text to illustrate the potential biological implications of these lipid changes. Suggested roles of sphingolipids were added in according to the comment (lines 322-330).
We believe that focusing on these key lipids provides a clearer narrative and a more manageable framework for understanding the initial findings. As we move forward with targeted studies, we intend to expand the biological context and investigate the roles of additional lipid species in early pregnancy. Thank you for your valuable feedback, and we hope this clarifies our approach.
- 5. Reporting elevated bile acid BA 24:1;O5 without discussing its specific role in pregnancy and embryonic development leaves the reader with unanswered questions.
Answer: Thank you for your comment regarding the elevated bile acid BA 24:1;O5. We appreciate the opportunity to clarify this point. We have included a detailed discussion of the biological relevance of BA 24:1;O5 in the manuscript. Specifically, we have highlighted its role in cholesterol metabolism and steroidogenesis, which are crucial processes during early pregnancy and embryonic development. These pathways are essential for the synthesis of steroid hormones that support the maintenance of pregnancy and the development of the embryo (lines 342-366).
We hope this addresses your concern and clarifies the biological significance of BA 24:1;O5 in the context of early pregnancy.
- 6. The paper severely lacks in translating its findings into actionable clinical or veterinary applications.
Answer: Thank you for your comment on the clinical and veterinary applications of our findings. We would like to emphasize that this study primarily aimed to explore the fundamental biology of lipidomic changes during early equine pregnancy, rather than to provide direct clinical interventions. Our goal was to generate new hypotheses regarding systemic changes associated with early gestation, which could be the basis for future research.
One significant observation from our study is the identification of systemic lipid changes occurring as early as day 7 post-ovulation. This finding represents a substantial shift from previous assumptions that such changes occur later in pregnancy. The implications of this are potentially far-reaching, suggesting that 1) early pregnancy diagnosis may be feasible with appropriate biomarkers, and 2) novel pregnancy-supportive interventions beyond progestins could be developed in the future. However, we are careful not to over-interpret these findings without further validation, as these hypotheses require additional research to establish their clinical relevance.
We believe that while this study provides a foundation for future investigations, it is important to acknowledge its exploratory nature and avoid drawing definitive conclusions until more evidence is available.
Based on your comment and our approach, we have added a few lines to the Discussion section to further elaborate on the need for future research to translate these preliminary findings into actionable clinical or veterinary applications. We hope this additional context clarifies the exploratory nature of our study and the potential implications of our results.
- 7. The discussion is notably weak in integrating these findings with the current body of literature on lipid metabolism in pregnancy.
Answer: We have strengthened the Discussion by integrating relevant literature on lipid metabolism in early pregnancy across species, highlighting how our findings contribute to the broader understanding of these processes (). This addition places our study within the context of existing knowledge and underscores its contribution to the field. (lines 348-354, 370-379, 389-395, 400-403)
- The authors need to go beyond merely identifying lipid changes and clearly articulate how these findings can be applied, especially as potential biomarkers for pregnancy diagnostics or reproductive management.
Answer: Thank you for your comment regarding the potential applications of our findings. We would like to highlight that this study was conducted as a preliminary investigation to explore systemic lipidomic changes during early equine pregnancy. The primary goal was to identify whether these changes occur early in gestation and to generate hypotheses for future research, rather than to establish definitive biomarkers for pregnancy diagnostics or reproductive management at this stage.
While we have identified several lipid alterations that could potentially serve as early indicators of pregnancy, further studies are needed to validate these findings in larger cohorts and to establish their reliability and specificity as biomarkers. Based on your feedback, we have added a few lines to the Discussion section to outline the future steps necessary to translate these preliminary results into clinically relevant applications (lines 430-433) . This will involve targeted lipidomic analyses and exploring the reproducibility of these changes across diverse populations of mares.
We appreciate your suggestion and have taken care to ensure that the limitations of the current study are clearly communicated, and that the findings are not over-interpreted without further evidence.
Overall:
Although the study clarifies the lipidomic alterations that occur during the early stages of equine pregnancy, it falls short in terms of offering a significant biological interpretation and a useful application of its findings. To increase the paper's significance, the authors must address these important problems. The study's value is still restricted in the absence of more thorough analysis, a larger cohort, and a more distinct link to therapeutic relevance.
Answer: Thank you for your constructive feedback. We acknowledge that while this study provides an initial insight into the lipidomic alterations associated with early equine pregnancy, it remains a preliminary investigation with certain limitations. Our primary aim was to establish a foundation for understanding these changes and to identify potential lipid biomarkers that could be further explored in future studies.
We recognize the need for a more comprehensive biological interpretation and the translation of these findings into clinically relevant applications. However, given the exploratory nature of this study and the relatively small cohort size, our focus was on generating hypotheses and identifying key lipid changes that warrant further investigation.

Reviewer 2 Report
Comments and Suggestions for Authors
The article under review aimed to investigate the dynamic changes in lipidomic profiles during the early stages of equine pregnancy, specifically focusing on days 7 and 14 post-ovulation. The study raises intriguing points, particularly regarding the method of lipid annotation. It remains unclear whether the authors acquired over 2000 high-quality fragmentation spectra for lipid identification. Clarification on this point, including the confidence level of lipid identification, is essential for manuscript clarity.
Several minor queries arise:
- What were the conditions of the sample from collection to centrifugation, i.e., over 1 hour? Is there information regarding lipid stability?
- "Metaboanalyst" should be capitalized.
- "Microsoft Excel 265" – should it be "365" instead?
- How confident are the authors in the position of unsaturated bonds?
- Has each identified lipid been confirmed through fragmentation?
- Fig. 1: Improvement in figure clarity is suggested.
- Please include a PCA plot. Additionally, provide validation results for each PLS-DA, including cross-validation and permutation tests.
- Instances referring to "proteins express" should be corrected.
- There are concerns about the appropriate usage of the term "lipid expression."
These points warrant attention to ensure the accuracy and comprehensibility of the manuscript.
Author Response
IJMS-3190061
RESPONSE TO REVIEWER COMMENTS
RESPONSE TO REVIEWER #2
The article under review aimed to investigate the dynamic changes in lipidomic profiles during the early stages of equine pregnancy, specifically focusing on days 7 and 14 post-ovulation. The study raises intriguing points, particularly regarding the method of lipid annotation. It remains unclear whether the authors acquired over 2000 high-quality fragmentation spectra for lipid identification. Clarification on this point, including the confidence level of lipid identification, is essential for manuscript clarity.
Answer: Thank you for your valuable feedback on the manuscript. We appreciate your recognition of the intriguing aspects of our study, particularly the dynamic changes in lipidomic profiles during early equine pregnancy. We have carefully reviewed your comments regarding lipid annotation and identification.
We understand the importance of providing clear and detailed information on the methods used for lipid identification, including the acquisition of high-quality fragmentation spectra and the confidence levels. In response, we have expanded the relevant sections of the manuscript to clarify our approach and ensure transparency in the reporting of our results.
We believe these revisions will enhance the clarity and robustness of the manuscript, addressing the points you raised. Thank you once again for your thoughtful review, and we hope the revised version meets your expectations.
- 1. Clarification on this point including the confidence level of lipid identification is essential for manuscript clarity.
Answer: We appreciate the reviewer nothing this oversight and we have added to our Methods section to detail the confidence levels of lipid identification, including the criteria used for lipid annotation and validation and notably the use of MS/MS fragmentation data to confirm lipid identities where applicable (please see lines 158 – 159).
- 2. What were the conditions of the sample from collection to centrifugation i.e., over 1 hour? Is there information regarding lipid stability?
Answer: We have added further details in our methods regarding the sample handling conditions from collection to centrifugation (please see lines 104 – 112). We also included a brief discussion on lipid stability, referencing relevant literature that supports the stability of lipids under our experimental conditions.
- 3. "Metaboanalyst" should be capitalized.
Answer: Thank you for noting this, we have ensured it is capitalized throughout. (line 166)
- 4. "Microsoft Excel 265" – should it be "365" instead?
Answer: We appreciate your attention to detail and have amended this.(Line 168)
- 5. How confident are the authors in the position of unsaturated bonds?
Answer: Building on the really helpful comment on lipid identification confidence, we have provided additional details around the confidence in the positioning of unsaturated bonds, including the use of high-resolution MS data and the challenges inherent in assigning double bond positions without additional standards or derivatization methods (please see lines 162 – 164).
- 6. Has each identified lipid been confirmed through fragmentation?
Answer: We have clarified in the Methods that all key lipids were confirmed through MS/MS fragmentation, providing supporting evidence in the supplementary materials for the lipid identities reported in the study. The lipids were identified using both CID and EAD fragmentation techniques using SCIEX ZenoTOF 7600 and lipid annotation was predicted using MSDIAL EAD library. (line 161-162)
- 7. Fig. 1: Improvement in figure clarity is suggested.
Answer: We have revised Figure 1 to improve its resolution, ensuring that the figure is legible and informative for the readers (attached as a fig in supplementary material)
- 8. Please include a PCA plot. Additionally, provide validation results for each PLS-DA including cross-validation and permutation tests.
Answer: We appreciate your suggestion to include a PCA plot; however, we believe that PCA may not be the most appropriate tool for our study, which focuses on biological samples from mares. Mares exhibit a wide range of variables, such as age, diet, genetics, and environmental factors, that go beyond pregnancy status. These factors introduce variability that is inherent in biological samples, making it challenging to apply standardization across all these variables. As discussed by Broadhurst and Kell (2006), PCA is an unsupervised method that primarily captures the largest sources of variance, which may not correspond to the biological questions of interest. Additionally, Westerhuis et al. (2008) highlight that supervised methods like PLS-DA are better suited for studies aiming to identify specific biomarkers, as they focus on maximizing class separation and are less affected by confounding variability.
Our objective is to identify lipid markers that are robust and can consistently indicate pregnancy across different mares, without needing to account for every variable individually. Rather than standardizing variables, which could complicate the study and limit its practical application, we aim to discover markers that naturally overcome these variabilities and reliably indicate pregnancy. This approach ensures that the findings are more broadly applicable and relevant in a real-world context, where such inherent differences among mares cannot always be controlled.
We have added a explanation to use PLS -DA in the Results section in line 180-186.
Broadhurst, D. I., & Kell, D. B. (2006). Statistical strategies for avoiding false discoveries in metabolomics and related experiments. Metabolomics, 2, 171-196.
Westerhuis, J. A., Hoefsloot, H. C., Smit, S., Vis, D. J., Smilde, A. K., van Velzen, E. J., ... & van Dorsten, F. A. (2008). Assessment of PLSDA cross validation. Metabolomics, 4, 81-89.
- 9. Instances referring to "proteins express" should be corrected.
Answer: We have corrected all instances to the appropriate terminology to “proteins” (line 206, 234, 259).
- 10. There are concerns about the appropriate usage of the term "lipid expression."
Answer: We have reviewed and refined the use of the term "lipid expression" throughout the manuscript to ensure that it accurately describes the changes in lipid levels observed in the study, avoiding any potential misinterpretation (lines 207, 235, 260, 268, 273).

Round 2
Reviewer 1 Report
Comments and Suggestions for Authors
The authors went through the feedback and have revised the manuscript as needed. In order to avoid drawing conclusions that are too strong, they have carefully reframed their findings and acknowledged the sample size limits. The study's exploratory aspect has been underlined, and the authors have reinforced the biological interpretation of important lipid alterations, especially the function of triglycerides and bile acids in the early stages of pregnancy. To support these conclusions, they have also included relevant research and suggested future study paths.
Overall, the paper has been significantly improved by the authors, and the study provides insightful information about lipidomic changes that occur during early equine pregnancy. It is therefore suggested that this manuscript be published.
Author Response
Comment: The authors went through the feedback and have revised the manuscript as needed. In order to avoid drawing conclusions that are too strong, they have carefully reframed their findings and acknowledged the sample size limits. The study's exploratory aspect has been underlined, and the authors have reinforced the biological interpretation of important lipid alterations, especially the function of triglycerides and bile acids in the early stages of pregnancy. To support these conclusions, they have also included relevant research and suggested future study paths.
Overall, the paper has been significantly improved by the authors, and the study provides insightful information about lipidomic changes that occur during early equine pregnancy. It is therefore suggested that this manuscript be published.
Answer: Thank you very much for your positive feedback and for acknowledging the revisions we made to the manuscript. We greatly appreciate your thoughtful comments, and we are pleased that the adjustments have helped improve the clarity and focus of our study. We are grateful for your recognition of the value of our study and the pathways it opens for future research. We look forward to the potential publication of our work and thank you again for your time and constructive suggestions throughout the review process.
Reviewer 2 Report
Comments and Suggestions for Authors
-
Thank you to the authors for their responses and the discussion. Please address the comments.
- Was the blood indeed kept on dry ice before centrifugation? Contact with such low temperatures would cause the sample to freeze and undergo hemolysis.
- I was not referring to the percentage confidence of lipid annotation, but rather the level, which may differ for each metabolite. This is discussed in many articles: “In metabolomics, five different levels exist, including the new ‘Level 0’ that requires the full 3D structure and stereochemistry information. More common are ‘Level 1’ annotations that are confirmed by two orthogonal parameters, such as retention time and MS/MS spectrum. etc.”
- I understand the argument regarding PLS-DA; however, since it is a supervised classification method, it has some drawbacks. As noted on the Metaboanalyst website, “When your sample size is small, it is better to use unsupervised methods (such as PCA) and simple but more robust statistical tests (such as t-tests or ANOVA), as supervised methods (such as PLS-DA) will be more susceptible to overfitting.” Therefore, if you wish to use PLS-DA, cross-validation must be performed. In the case of PCA, when there is a significant influence from other factors on variability, we look for principal components (PCs) that explain the variability we are interested in.
Author Response
RESPONSE TO REVIEWER #2
- 1. Was the blood indeed kept on dry ice before centrifugation? Contact with such low temperatures would cause the sample to freeze and undergo hemolysis.
Answer: Thank you for pointing that out, and we apologise for the oversight. It was mistakenly written as dry ice. The blood samples were in fact kept on normal ice. We have now corrected this in the manuscript (line 105).
- 2. I was not referring to the percentage confidence of lipid annotation, but rather the level, which may differ for each metabolite. This is discussed in many articles: “In metabolomics, five different levels exist, including the new ‘Level 0’ that requires the full 3D structure and stereochemistry information. More common are ‘Level 1’ annotations that are confirmed by two orthogonal parameters, such as retention time and MS/MS spectrum. etc.”
Answer: Thank you for further clarifying this point. We can confirm that the lipid identifications in our study are at Level 1, as defined by a seven-level classification system for lipid annotation outlined by Fernández et al, 2024 Commun Biol. This means that the annotations were confidently made using two different fragmentation techniques, collision induced dissociation (CID) and electron activation dissociation (EAD), which provides an additional information of the regio-isomer and double bond positions of the lipid structure. We have used MS-DIAL silico predicted lipid library based on both CID and EAD fragmentations for identification of the lipid species. We have explicitly stated this in the manuscript (lines 158 – 163), ensuring that the identification level is clearly communicated.
- 3. I understand the argument regarding PLS-DA; however, since it is a supervised classification method, it has some drawbacks. As noted on the Metaboanalyst website, “When your sample size is small, it is better to use unsupervised methods (such as PCA) and simple but more robust statistical tests (such as t-tests or ANOVA), as supervised methods (such as PLS-DA) will be more susceptible to overfitting.” Therefore, if you wish to use PLS-DA, cross-validation must be performed. In the case of PCA, when there is a significant influence from other factors on variability, we look for principal components (PCs) that explain the variability we are interested in.
Answer: Thank you for your insightful feedback regarding the use of PLS-DA. We fully acknowledge the potential drawbacks of using a supervised classification method, particularly with a small sample size, as it can lead to overfitting. In light of this, we have carefully performed cross-validation as part of the PLS-DA analysis, and the results are presented in the manuscript (Supplementary Table ST2.1). While the high R² values suggest a good fit, the lower Q² values reflect the limitations in predictive power, supporting your concern about overfitting. However, despite these limitations, we have identified several important lipids that contribute to the separation between the NP and P groups as well as PD7 and PD14, as indicated by the VIP scores (Supplementary Table ST2.2). We have acknowledged the limitations of the PLS-DA (line 180 – 194).

Round 3
Reviewer 2 Report
Comments and Suggestions for Authors
The results of the PLS-DA analysis should be removed. It is suggested to either use PCA with the appropriate combination of PCs on the axes or only standard statistical analyses
Author Response
RESPONSE TO REVIEWER #2
- 1. The results of the PLS-DA analysis should be removed. It is suggested to either use PCA with the appropriate combination of PCs on the axes or only standard statistical analyses
Answer: Apologies for the confusion over our response and we appreciate you taking the time to highlight this for us. Accordingly, we have removed the PLS-DA and focused solely on the statistical comparisons already discussed through the volcano plots. We have adjusted the manuscript throughout to remove any instances of data relating to the PLS-DA analysis (lines 176 – 198, 212 – 216, and removal of Figure 1) and have readjusted the figure numbers, as highlight in yellow throughout.